# Population Dynamics and Yield Loss Assessment for Pea Aphid, *Acyrthosiphon pisum* (Harris) (Homoptera: Aphididae), on Lentil in Morocco

**DOI:** 10.3390/insects12121080

**Published:** 2021-11-30

**Authors:** Karim El Fakhouri, Abdelhadi Sabraoui, Zakaria Kehel, Mustapha El Bouhssini

**Affiliations:** 1International Center for Agricultural Research in the Dry Areas (ICARDA), Entomology Laboratory, Rabat Institutes, Rabat P.O. Box 6299, Morocco; sabraoui92@gmail.com; 2International Center for Agricultural Research in the Dry Areas (ICARDA), Genetic Resources Program, Rabat Institutes, Rabat P.O. Box 6299, Morocco; z.kehel@cgiar.org; 3AgroBioSciences Research Division, Mohammed VI Polytechnic University, Lot 660, Hay Moulay Rachid, Ben Guerir P.O. Box 43150, Morocco; Mustapha.ElBouhssini@um6p.ma

**Keywords:** pea aphid, lentil, population fluctuations, climatic factors, avoidable yield loss

## Abstract

**Simple Summary:**

In Morocco, lentil is being grown under rainfed conditions and plays an important role in cereal-based cropping systems. Pea aphid (*Acyrthosiphon pisum* Harris) is one of the most important constraints limiting the yield of lentil in several countries; however, the extent of yield loss it causes in Morocco is unknown. Pea aphid weakens the plant directly by sucking its sap, and can also spread viruses from infected plants to healthy ones. Currently, there are no effective tools available for *A. pisum* control, with most farmers being reliant on chemical insecticides. The aim of this study is to investigate the population fluctuation of pea aphid over different seasons and their effects on yield loss in Morocco. Correlation analysis was performed to find out the extent of influence of weather parameters on the population dynamics of an aphid population over different seasons. The results demonstrated that the avoidable losses due to aphid infestation were in the range of 4.56–12.51%. The pea aphid populations increased rapidly between March and April when climate factors and the plants became more suitable for aphid development. The maximum temperature, relative humidity, and wind speed influenced pea aphid infestation on different lentil varieties.

**Abstract:**

Pea aphid (*Acyrthosiphon pisum* Harris) is the major insect pest of lentil in Morocco. We investigated pea aphid mean numbers and yield losses on three lentil varieties at one location during three successive cropping seasons during 2015–2018. The effects of several weather factors on pea aphid population dynamics were investigated. Population density increased in early spring followed by several peaks during March–April and then steeply declined during the late spring. Aphid populations peaked at different times during the three years of the study. In 2016, higher populations occurred during the second and third weeks of April for Abda and Zaria varieties with averages of 27 and 28 aphids/20 twigs, respectively. In 2017, higher populations occurred on the 12th and 13th standard meteorological weeks (SMWs) for Zaria with averages of 24.7 and 27.03 aphids/20 twigs, respectively. In 2018, the population peaked for all varieties at three different times, 11th, 13th, and 17th SMW, with the highest for Zaria being 26.00, 47.41, and 32.33 aphids/20 twigs. Pea aphid population dynamics changed with weather conditions. The number of aphids significantly and positively correlated with maximum temperature, but significantly negatively correlated with relative humidity and wind speed. The minimum temperature and rainfall had non-significant correlations. Pea aphid infestation resulted in losses of total seed weight for all lentil varieties, with the highest avoidable losses for Bakria being 12.51% followed by Zaria with 7.72% and Abda with 4.56%. These losses may justify the development of integrated management options for control of this pest.

## 1. Introduction

Lentil (*Lens culinaris* Medikus) is one of the most important annual food legumes in the world, particularly in three major distinct agro-ecological zones: Mediterranean, sub-tropical savannah, and northern temperate [1]. In many countries, lentil crops play important roles in human nutrition and nutritional security, straw for animal feed, and soil fertility improvement. Lentil seeds provide an affordable source of dietary protein (content of 22–35%), a low glycemic index, and a good source of essential amino acids, minerals, fiber, antioxidants, and carbohydrates [2]. Lentil crops provide many agronomic benefits when grown in rotation with cereals and other crops, improving soil fertility through biological nitrogen fixation [3].

The most recent available data show that globally in 2017, 7.5 million metric tons of lentils were produced in 6.5 million hectares of land for an average yield of 1.13 t/ha [4]. Lentils are the fourth most-commonly harvested grain legume after soybeans, kidney beans, and peas. Canada is the world’s leading producer of lentils with 2.7 million ha and a gross yield of 3.73 million metric tons/ha, followed by India and Turkey [4]. In Morocco, lentil is ranked the third most-important food legume after faba bean and chickpea, with 41,000 ha and annual production averaging 28,000 metric tons during the past 10 years [4]. The average yield of lentil in Morocco was 43% below the world average of 1.09 t/ha during the period 2008–2017 [4].

Similarly to many other crops that grow in North Africa and semi-arid zones, lentil production often faces several abiotic limiting factors, such as terminal drought stress, mineral imbalances, high temperatures during pod filling, and cold temperatures in winter at high elevations. Among biotic stresses, insect pests are one of the most important constraints limiting yield. About three dozen insect pests of various genera and species have been reported to infest lentil under field and storage conditions across different regions of the world [5]. However, few species cause sufficient yield losses to be of economic significance and to require control measures. The field insect pests include aphids (*Aphis craccivora* Koch and *Acyrthosiphon pisum*), Sitona weevils (*Sitona crinitus* Herbst), Lygus bugs (*Lygus* spp.), cutworms (*Agrotis ipsilon*), thrips (*Kakothrips* and *Frankiniella*), bud weevils (*Apion arrogans* Wenck), and pod borer (*Helicoverpa armigera* Hubner). During storage, several species of seed beetles including *Bruchus* spp. and *Callosobruchus* spp. can cause severe damage [6]. 

The pest status of each insect varies greatly among regions. Crop loss may depend upon species characteristics, landscape context, and patch size [7]. Pea aphid, *Acyrthosiphon pisum* Harris (Hemiptera: Aphididae), was first observed in Europe, from where it spread worldwide under temperate climate and now has a cosmopolitan distribution [8]. The pea aphid is an oligophagous herbivore that causes direct damage to plants by sucking sap from the tender leaves, stems and pods. It mostly colonizes young leaves and growing points and, more seriously, transmits plant viruses. The aphid infestation results in a reduction in vegetative growth, yellowing, and stunting of plants, with yield reduction achieved through decreased seed formation and seed size [6,9]. A number of authors have studied the impact of abiotic factors on pea aphid density, but the results were varied. In general, the environmental variations are also important variables for insect survival, behavior, abundance, and distribution, with temperature being the most dominant variable [10]. According to different studies, temperature had a positive and significant effect on pea aphid population, while humidity and wind speed played a negative role. Other studies related changes in pea aphid densities to planting dates [11,12].

Currently, there are no effective non-chemical tools available for pea aphid management; most farmers rely on chemical insecticides. Pea aphid is a limiting factor in lentil production in several countries and is the most abundant pest of lentil in Morocco; however, the extent of yield loss that it causes in Morocco is unknown.

The present study investigated the population fluctuation of pea aphid over different seasons and their effects on yield loss in Morocco.

## 2. Materials and Methods

### 2.1. Study Site and Experimental Design

The experiments were conducted at ICARDA’s Marchouch Research Station, Morocco (33°61′ N and 6°71′ W, with altitude of 410 m) over three cropping seasons (2015–2018) to study the population dynamics of aphids and their impact on the lentil. They were laid out following a split plot design with three replications using three Moroccan lentil varieties: Abda, Zaria, and Bakria. Sowing occured on 27 November 2015, 19 November 2016, and 9 December 2017. The main plot was considered as an insecticide treatment factor, and the sub-plot represents different lentil varieties. The unit sub-plot consisted of eight rows of 4 m length with a row spacing of 30 cm and seeding rate of 50 kg/ha. The soil texture was clay and clay to calcareous, and the recommended agronomic practices and irrigation regimes were adopted. The control plots were kept protected from aphid infestation by applying the selective carbamate insecticide pirimicarb (Pirimor 50 DG, Syngenta, Morocco) at 75 g/hL once a week, using a low-pressure backpack sprayer. The spaces between the blocks and plots were 4 m in order to avoid any drift from adjacent plots. In addition, to avoid insecticide drift to other plots, polyethylene sheets were used during spraying to protect unsprayed plots.

### 2.2. Field Monitoring

At seven-day intervals, pea aphids were randomly sampled per 20 twigs of 10 cm in length during the morning hours of 8:30–11:00 a.m. using a beating sheet from the two-side rows of each plot from the untreated plot, avoiding boarder rows of each plot from the untreated split of each block. The aphid population was counted with the help of magnifying lens. In each season, plots were monitored starting from seedling establishment to determine the start of infestation by pea aphids. The central four rows were kept undisturbed to record grain yield. The aphids were collected starting from their first appearance at the early to mid-vegetative stage of the plants until harvest.

### 2.3. Yield, Avoidable Yield Loss, and Yield Increase over Untreated Control

The central four rows were kept undisturbed to record grain yield. At maturity, these four central rows of each plot of both treated and untreated splits were harvested and converted to record seed yield in kg/ha. The difference between the weight of grain yield in protected and unprotected plots was considered as losses.

The percentage increase in yield was calculated using the following formula.
$$\frac{P1-P2\;}{P2}$$

The percentage avoidable loss in yield due to sucking pests was calculated using the following formula [13]:$$\mathrm{Percentage}\;\mathrm{avoidable}\;\mathrm{loss}=\frac{P1-P2\;}{P1}$$
where P1 is mean yield of protected plots (kg/ha), and P2 is mean yield of unprotected plots (kg/ha).

### 2.4. Data Analyses

The repeated measurements (number of pea aphid over weeks) were analyzed using linear mixed models and using the Asreml-R package [14]. Year, varieties, and their interaction together with the within year interaction between varieties and weeks were considered to be fixed. For each year, a between weeks variance-covariance matrix was assessed while modelling the data to reflect the correlation between weeks. An autoregressive correlation structure order 1 (AR1) was used. Within year variation due to blocking was fitted as random effects in the linear mixed model. In each year, the estimates of lines in each week were then extracted from the model using the best linear unbiased estimators (BLUEs). The means of seed weight of each year were separated by Student–Newman–Keuls test at *p* < 0.05 using Genstat Version 19. 

The Pearson correlation coefficients (PCCs) were computed between mean over lines of weekly number of aphids (N = 8, 15, 11 weeks in 2016, 2017, and 2018, respectively) and the weekly meteorological factors (maximum and minimum temperature, relative humidity, total rainfall, and wind speed) using Genstat Version 19. The agrometeorological data were obtained by using the web platform FieldClimate.com. In order to evaluate yield loss and grain yield-pea aphid density relationship, individual yields and the maximum aphid density for each plot were identified using the Pearson coefficient, where yield for each plot (measured in kg per ha) was the dependent variable and the maximum aphid density (measured in aphids per leaf) was the independent variable [15]. The maximum aphid density was defined as the highest average plot density per leaf recorded across all sampling weeks for each variety-year. This measure was used to estimate the yield-aphid density relationship. The percentage yield loss for each plot was determined by using the following equation: percentage yield loss = (yield of treated variety − yield of untreated variety) × 100/yield of treated variety.

## 3. Results

### 3.1. Population Dynamics of Pea Aphid

#### 3.1.1. 2016. Cropping

During 2016, the main effects of variety and the interaction of variety and sampling dates (weeks) were highly significant (F = 72.86; NumDF = 21; DenDF = 1274.3; *p* < 0.001) (Table 1). Pea aphids first appeared in the unprotected plots during the last week of March, but the protected plots remained free of any aphid infestation. The pea aphid population increased during April and peaked during the second and third week of April (16th and 17th standard meteorological weeks (SMWs)) when the crop was at full bloom stage, at mean temperatures of 16.45 °C and 17.22 °C, and relative humidity of 71.78% and 77.07%, respectively. Higher aphid populations were observed for varieties Abda and Zaria with averages of 27 and 28 aphids/20 twigs on 16 and 23 April, respectively. The aphid population reached its peak for the Bakria variety during the same period (16th and 17th SMW) with averages of 24 and 25 aphids/20 twigs, respectively (Figure 1A). A total of 7023 aphids/20 twigs was counted for Abda during the entire season, followed by Zaria with 6690 and Bakria with 5747 aphids/20 twigs. From the last week of April, aphid numbers decreased considerably and had completely disappeared from the third week of May at full pod cavity stage for all lentil varieties tested.

#### 3.1.2. 2017 Cropping

During 2017, the main effects of variety and the interaction between variety × sampling dates (weeks) were highly significant (F = 47.05; NumDF = 27; DenDF= 1687.5; *p* < 0.001) (Table 1). Pea aphid patterns by the end of January were at a vegetative stage (i.e., earlier than in 2016). The reasons for this may possibly be due to early planting compared to 2015 (8 days earlier) and 2017 (20 days earlier). A second reason is probably due to the suitable conditions, which mostly consists of early rain recorded in October and November of 2016, allowing the plants to become more suitable for aphid development by the end of January. In addition, the mild winter during 2016 allowed the high survival of over-wintering nymphs and adults and, thus, caused the early infestation of pea aphids on lentil varieties. Aphid density gradually increased during March, and the first peak was recorded for Bakria during the second week of March (10th SMW) at full bloom stage, with an average of 23 aphids/20 twigs, a mean temperature of 16.39 °C, and relative humidity of 79.4%. The second population peak was on 8 and 15 April (14th and 15th SMW) at full flowering and early-pod developmental stages, with a mean population of 24 and 24.5 aphids/20 twigs, respectively, with corresponding mean temperature of 16.56 °C and 19.48 °C and relative humidity of 61.44% and 71.47%.

The peak numbers of aphids for varieties Zaria and Abda lasted from 25 March to 1 April (12th to 13th SMW) at the full bloom stage at a mean temperature of 11.51 °C and 14.94 °C and relative humidity of 84.74% and 79.24%, respectively. The highest peak was for Zaria with an average of 24.7 and 27.03 aphids/20 twigs, respectively; and the corresponding lowest population was for Abda with 15.06 and 17.56 aphids/20 twigs (Figure 1B). A seasonal total of more than 12,530 aphids were counted on Zaria, followed by 10,282 on Bakria and 7674 on Abda.

From early April, the aphid numbers decreased considerably for Abda and Zaria and from the last week of April for Bakria. Aphids completely disappeared from mid-May for all varieties at full pod cavity stage.

#### 3.1.3. 2018 Cropping

During 2018, the main effects of the variety and the interaction between variety × weeks were highly significant (F = 117.5; NumDF = 30; DenDF = 1704; *p* < 0.001) (Table 1). The population fluctuation of pea aphid was characterized by several peaks during the 2018 season. Initially, the pest appeared in early March at vegetative and early bloom stages, and the population then increased and peaked for all varieties at three different times on 15 March, 29 March, and 26 April at full flowering and early-pod developmental stages, at mean temperatures of 12.7 °C, 13.36 °C, and 15.53 °C, respectively, and a corresponding relative humidity of 90.37%, 86.65%, and 92.30%. The late peaks recorded at the last week of April could be related to the late planting date (second week of December) of all varieties compared to the previous seasons. The highest aphid population was for the Zaria variety with 26.00, 47.41, and 32.33 aphids/20 twigs; the peak for Bakria was 32.51, 32.15, and 16.25 aphids/20 twigs; and for Abda it was 22, 14.86, and 13.48 aphids/20 twigs for 11th, 13th, and 17th SMW, respectively (Figure 1C). 

The highest total number of pea aphids was 13,597 on Zaria, followed by Bakria with 8711, and Abda with 5926. The aphid numbers decreased considerably for all varieties until their disappearance by the end of May.

A repeatedly measured ANOVA showed that the year effect was not significant on the mean number of pea aphid (Table 1). Pea aphid population levels follow a consistent pattern over the years, whereas the interaction between year × variety was highly significant (F = 27.18; df = 4; *p* < 0.001). 

### 3.2. Yield Loss Caused by Pea Aphid

The main effects of variety (F = 343.1; NumDF = 2; DenDF = 10.6; *p* < 0.001) and year (F = 3953; NumDF = 3; DenDF = 4.6; *p* < 0.001) and the variety × year interaction were highly significant (F = 33.06; NumDF = 4; DenDF = 13.4; *p* < 0.001) (Table 2). In the 2015–2016 season, the interactions between variety × treatment insecticide were highly significant (F = 75.26; NumDF = 3; DenDF = 10; *p* < 0.001); higher seed yield was recorded in protected compared to unprotected plots. Grain yields both in protected and unprotected lentil varieties and their respective avoidable yield loss and yield increase due to aphids are presented in Table 3. There was a higher mean seed yield of 1235 kg/ha for protected plots of Abda, followed by Bakria with 1112 kg/ha. Both Abda and Bakria showed losses due to pea aphid, with maximum yield losses of 149 and 109 kg/ha, respectively. The highest yield increase in protected over unprotected plots was 13.72% for Abda followed by Bakria with 10.86%. The avoidable losses due to aphid infestation ranged from 9.80 to 12.06% for Bakria and Abda, respectively; no yield loss was observed on Zaria.

In 2016–2017, ANOVA showed highly significant interaction between variety × treatment insecticide (F = 6.65; NumDF = 3; DenDF = 12; *p* =0.006). However, there were no differences in seed yield between protected and unprotected plots for both varieties Bakria and Zaria. The maximum losses in grain yield were for Bakria and Zaria with 194 and 102 kg/ha, respectively. The highest yield increase over control was obtained for Bakria with 10.26% followed by Zaria with 5.85%. The avoidable yield losses due to pea aphid on lentil varieties ranged from 5.53% on Zaria to 9.31% for Bakria; no yield loss was observed on Abda during this season.

In 2017–2018, there were highly significant interactions between variety x treatment insecticide (F = 7.918; NumDF = 3; DenDF = 7.6; *p* = 0.01). The analysis showed a highly significant difference in seed yield between treated and untreated plots, except for Abda variety. Grain yield was higher for all varieties compared to the previous seasons, probably because the 2017–2018 season received more rainfall and, thus, was favorable for lentil production. Among the lentil varieties tested, the maximum yield loss was for Bakria (649 kg) followed by Zaria (601 kg) and Abda (64 kg). The highest yield increase over control was 22.61% and 21.41% for Bakria and Zaria, respectively. The avoidable yield losses were 1.62%, 17.64%, and 18.44% for Abda, Zaria, and Bakria, respectively. 

The results revealed that the grain yield and percent loss in grain yield was significantly associated with the aphid population (Table 4). The grain yield per hectare was significantly negatively correlated with the aphid population on different lentil varieties for different seasons. However, a non-significant negative correlation was found between aphid population and grain yield for Abda in 2017 and 2018. The correlation was non-significant and positive for Zaria in 2016. 

However, there was significant and positive correlation between yield loss and aphid density on different lentil varieties for different seasons, except for the Abda variety in 2016, which revealed that high infestation of pea aphids might cause great yield losses in lentil.

### 3.3. Effect of Climatic Factors on Pea Aphid Population

During 2015–2016, the population of pea aphid on all lentil varieties increased rapidly with crop growth and the rise in temperature. There was a significant positive correlation between maximum temperature and aphid population for Bakria, Abda, and Zaria (r = 0.62, 0.70 and 0.69, *p* < 0.05, respectively) (Table 5). There was a significant increase in pea aphid density starting from the second week of April (15th SMW) at the maximum temperature of 20.75 °C. However, a non-significant positive correlation was found between minimum temperature and aphid population for all varieties. The lowest temperature of early spring (7.95–14.31 °C) was not significantly correlated with the increase in the aphid population, but mean relative humidity was significantly negatively correlated with the aphid population for Abda and Zaria (r = −0.70 and −0.68, *p* < 0.05, respectively). The other weather parameters, both rainfall and wind speed, were negatively but non-significantly correlated with the aphid population.

During 2016–2017, the maximum temperature was significantly positively correlated with the aphid population on Bakria (r = −0.52, *p* < 0.05), but this was non-significant for Abda and Zaria (Table 6). The pea aphid numbers declined with the increase in wind speed. The correlation of wind speed with aphid populations was significant and negative on Abda and Zaria (r = −0.5, *p* < 0.05 and r = −0.53, *p* < 0.05, respectively). Additionally, minimum temperature, mean relative humidity, and total rainfall were negatively correlated with aphid populations but were non-significant.

During 2017–2018, the maximum temperature, minimum temperature, and total rainfall were non-significantly negatively correlated with aphid population. The mean relative humidity and wind speed were positively but non-significantly correlated with the pea aphid population (Table 7). Thus, during 2017–2018, all weather parameters were not significantly related to aphid abundance for all varieties. Seasonal variations in the weather factors affected the development and trend in population fluctuations of pea aphid on the different lentil varieties, explaining the multiple peaks of infestation during this season.

## 4. Discussion

Pea aphid population dynamics revealed significant variations for different thermal environments and seasons (Figure 1). The population of Pea aphid follow a consistent pattern over the years, with several peaks recorded on all lentil cultivars from the second week of March to late April of each year. The pea aphid population density increased in early spring followed by several peaks during both March and April and then declined steeply during the late spring. Through the three seasons of field monitoring, pea aphid population peaks covered the early flowering period to early pod setting period. The present findings are consistent with those of [9,16], who reported that pea aphid infestation can reach 100% at peak flowering and at early pod setting of lentil, causing total crop failure in Ethiopia from time to time.

In Ethiopia, pea aphids multiply in large numbers in early September through to January [17] when minimum temperatures are low. In the present study, aphid populations were influenced by climatic factors, with a significant positive correlation with maximum temperature and negative correlations with relative humidity and wind speed. Increasing the maximum temperature promoted aphid population increase while increased minimum temperature, rainfall, and relative humidity suppressed it.

The study of [11] showed similar results of aphid population increase when maximum temperature increased, and rainfall reduced on field pea (*Pisum sativum* L.) in Ethiopia. This is consistent with [18] who reported a positive correlation between maximum temperature and pea aphid populations and negative correlations for minimum temperature and relative humidity on field pea in Ethiopia. In contrast, the daily nymph production of pea aphid was significantly and positively correlated with the minimum temperature on lentil in a greenhouse [12]. A previous study showed that under field conditions, pea aphid required an average of 12.3 days above 5.56 °C to produce live offspring [19]. Laboratory studies provided estimates of both lower (2.73 °C) and upper (26.02 °C) thresholds for nymphal development rate of pea aphid [20]. The optimum temperature for pea aphid in general is suggested to be 18–24 °C [21]. Prolonged exposure to high temperatures, such as 25 °C or above is detrimental to aphid development and survival [22]. However, exposure to short periods of high temperatures lasting for a few hours can also critically affect aphid species and their interactions with natural enemies [23,24]. In the present study, wind velocity decreased aphid numbers in all varieties tested. In a recent study, Follman et al. [25] also reported that pea aphids were negatively influenced by mechanical stimulation (wind and plant contact) that was indirectly mediated by their host plants. Pea aphids had reduced fecundity when placed on plants previously exposed to mechanical stimulation from either leaf-to-plant contact or wind from a fan. Ben-Ari et al. [26] showed that pea aphid displays a specific crouched body posture, which reduces their chance of being carried off from the plant by sudden winds. Direct exposure to wind can have indirect effects on prey by altering predator behavior and can alter insect dispersion [26,27].

In addition, the average relative humidity was significantly negatively related to the pea aphid population during 2015–2016 for Abda and Zaria. This is consistent with the findings of Wale [18], who found that increasing relative humidity suppressed pea aphid populations on field pea in Northwestern Ethiopia.

The results from the present investigation suggested that percent loss in grain yield was significantly associated with aphid population. A negative correlation between grain yield per hectare and aphid population, which revealed that high infestation of aphids might cause great yield losses in lentil. In addition, a positive correlation was reflected between yield loss and aphid density. These findings are in agreement with those of Paudel et al. [8] where a significant relationship was detected between pea aphid density and relative economic yield for the plants infested during the reproductive stage (45 days after emergence).

In the present study, the highest avoidable losses due to aphid infestation was for the Bakria variety with 12.51% followed by Zaria with 7.72% and Abda with 4.56%. This difference in yield losses could be expressed by different levels of tolerance exhibited by each variety relative to pea aphid. In a recent study, Geteneh et al. [28] showed that the expression of the three modes of resistance (antibiosis, antixenosis, and tolerance) to pea aphid differed in released lentil varieties compared to susceptible accessions.

The current research finding was consistent with avoidable yield loss due to *Aphis craccivora* Koch infestation recorded on lentil in Bangladesh, with loss ranging within 0.9–9.0% depending on sowing date [29]. In Ethiopia, the pea aphid causes higher yield losses on lentil, with a range of 4–72% and an average of 30%. This depends on several factors such as planting date, season, production system, and location [30,31]. Perez Andueza et al. [32] reported that several key pests of lentil in central Spain, including aphids (pea aphid and *Aphis craccivora*), resulted in average seed weight losses of 12–20%.

Pea aphids damage lentil plants directly by sucking plant sap and more seriously by transmitting economically important viruses. The pea aphid is a known vector of a number of viruses affecting legume field crops, with all but two being stylet-borne [33,34].

## 5. Conclusions

We showed that pea aphid numbers rose to high levels in March and April when climate factors and plants become more suitable for aphid development and are mostly at full flowering and early-pod developmental stages. Abiotic factors such as maximum temperature, relative humidity, and wind speed influenced pea aphid infestation on different lentil varieties. The results also concluded a negative correlation between grain yield per hectare and aphid population, which revealed that high infestation of aphids might cause great yield losses in lentil. The avoidable losses due to aphid infestation were in the range of 4.56–12.51%. These losses justify the development of integrated management options for the control of this pest in Morocco.

## Figures and Tables

**Figure 1 insects-12-01080-f001:**
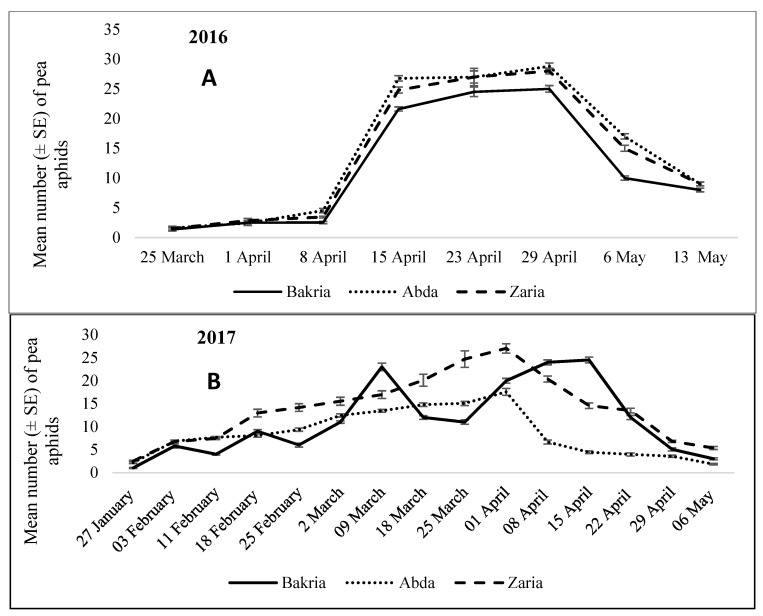
Population of pea aphids per 20 twigs for three lentil varieties during 2016 (**A**), 2017 (**B**), and 2018 (**C**) (means ± SE of 20 samples).

**Table 1 insects-12-01080-t001:** The repeated measures ANOVA examining the effect of variety and year and the interaction of sampling date (weeks) × variety in each year on mean number of Pea aphid on Lentil in Marchouch.

Factors	NumDF	DenDF	F.inc	Pr
(Intercept)	1	2.1	6957	<0.001
Year	2	2.9	2.38	0.2425
Variety	2	3639.3	170.1	<0.001
Year × Variety	4	2068	27.18	<0.001
Variety × Weeks in 2016	21	1274.3	72.86	<0.001
Variety × Weeks in 2017	27	1687.5	47.05	<0.001
Variety × Weeks in 2018	30	1704	117.5	<0.001

NumDF = numerator degrees of freedom; Dendf = denominator degrees of freedom.

**Table 2 insects-12-01080-t002:** ANOVA describing the interaction effect of variety and year on mean seed yield on Lentil in Marchouch Station.

Factors	NumDF	DenDF	F.inc	Pr
Years	3	4.6	3953	<0.001
Varieties	2	10.6	343.1	<0.001
Years × Varieties	4	13.4	33.06	<0.001
Varieties × Treatment in 2016	3	10	75.26	<0.001
Varieties × Treatment in 2017	3	12	6.65	0.006
Varieties × Treatment in 2018	3	7.6	7.918	0.01

NumDF = numerator degrees of freedom; DenDF = denominator degrees of freedom.

**Table 3 insects-12-01080-t003:** Effect of aphid infestation on yield, avoidable yield loss, and yield increase due to aphid protection on different varieties at Marchouch station during three cropping seasons: 2015–2016, 2016–2017, and 2017–2018.

Varieties	Seed Weight (kg/ha)			Yield Loss (kg/ha)			Avoidable Yield Loss (%)			Yield Increase Over Control (%)		
	2015–2016	2016–2017	2017–2018	2015–2016	2016–2017	2017–2018	2015–2016	2016–2017	2017–2018	2015–2016	2016–2017	2017–2018
Bakria	1003 b	1889 a	2870 a	109	194	649						
Bakria treated	1112 c	2083 a	3519 b				9.80	9.31	18.44	10.86	10.3	22.6
Zaria	973 b	1741 a	2806 a	0	102	601						
Zaria treated	892 a	1843 a	3407 b				0	5.53	17.64	0	5.85	21.4
Abda	1086 c	2880 c	3880 b	149	0	64						
Abda treated	1235 d	2380 b	3944 b				12.06	0	1.62	13.72	0	1.64
CV (%)	10.94	19.75	14.48				-	-	-	-	-	-
SEM	27.00	48.80	55.70									

Means followed by the same letter(s) within a column do not significantly differ at *p* = 0.05; CV is coefficient of variation; SEM: standard error of the mean.

**Table 4 insects-12-01080-t004:** Correlated response of pea aphid population with yield loss and grain yield/hectare.

Population	Season	Variety	Grain Yield (kg/ha)	Losses in Grain Yield (%)
No. of pea aphids causing yield losses	2016	Bakria	−0.998 *	0.401 *
	2017		−0.124 *	0.575 *
	2018		−0.943 **	0.940 *
	2016	Abda	−0.922 *	0.675 *
	2017		−0.627 ns	
	2018		−0.953 ns	0.923
	2016	Zaria	0.838 ns	
	2017		−0.014 *	0.900 *
	2018		−0.466 **	0.893 **

* Significant at 5% probability level; ** significant of 1% probability level; ns—not significant.

**Table 5 insects-12-01080-t005:** Influence of weather factors on mean number of pea aphid, *A. pisum*, per 20 twigs of three lentil varieties during 2015–2016.

		Abiotic Factors					Mean no. of Pea Aphid per 20 Twigs in the Tested Lentil Varieties		
SMW	Date and Month of 2016	Maximum (°C)	Minimum (°C)	Mean RH (%)	Total Rainfall (mm)	Wind Speed (m/s)	Bakria	Abda	Zaria
12	19–25 March	16.62	7.95	86.86	2.40	1.50	1.37	1.62	1.52
13	26 March to 1 April	19.77	7.66	84.81	2.40	0.84	2.50	2.43	2.85
14	2–8 April	18.86	7.80	82.20	0.51	1.34	2.55	4.55	3.46
15	9–15 April	20.75	8.38	78.25	0.00	1.19	21.61	26.76	24.81
16	16–22 April	22.92	10.16	71.78	1.40	1.19	24.50	27.00	27.00
17	23–29 April	24.34	10.09	77.09	1.20	0.83	25.00	28.8	28.00
18	30 April to 6 May	25.98	14.31	68.52	0.80	1.17	10.00	17.00	15.00
19	7–13 May	19.70	11.05	81.34	0.60	2.16	8.00	9.00	9.00
Seasonal mean		21.12	9.68	78.86	1.16	1.28	7.68	8.91	8.44
Correlation Coefficient (r) between Mean Aphid Population and Abiotic Factors									
Maximum temperature (°C)							0.62 *	0.70 *	0.69 *
Minimum temperature (°C)							0.24	0.35	0.33
Mean relative humidity (%)							−0.61	−0.70 *	−0.68 *
Total rainfall (mm)							−0.37	−0.45	−0.41
Wind speed (m/s)							−0.38	−0.36	−0.37

SMW, standard meteorological week; RH, relative humidity; * significant at *p* < 0.05.

**Table 6 insects-12-01080-t006:** Influence of weather factors on mean number of pea aphid, *A. pisum*, per 20 twigs of three lentil varieties during 2016–2017.

		Abiotic Factors					Mean no. of Pea Aphid per 20 Twigs in the Tested Lentil Varieties		
SMW	Date and Month of 2017	Maximum (°C)	Minimum (°C)	Mean RH (%)	Total Rainfall (mm)	Wind Speed (m/s)	Bakria	Abda	Zaria
4	22–28 January	14.41	4.98	84.98	15.60	1.41	1.01	2.16	2.51
5	29 January to 4 Feb	18.15	6.41	87.23	0.80	0.84	5.78	6.85	6.76
6	5–11 February	17.99	5.59	85.60	6.20	1.39	4.00	7.66	7.43
7	12–18 February	19.26	7.14	83.84	1.20	0.84	9.00	8.08	12.96
8	19–25 February	14.56	9.22	98.63	0.00	0.40	6.00	9.50	14.16
9	26 February to 4 March	17.62	6.76	91.14	0.00	0.66	11.00	12.45	15.53
10	5–11 March	24.44	8.33	79.40	0.00	0.34	23.00	13.45	16.96
11	12–18 March	18.55	6.98	85.93	0.40	0.67	12.00	14.78	20.11
12	19–25 March	16.65	6.36	84.74	10.60	0.94	11.00	15.06	24.70
13	26 March to 1 April	22.59	7.28	79.24	0.60	0.81	20.00	17.56	27.03
14	2–8 April	26.26	6.85	61.44	0.40	0.90	24.00	6.66	20.38
15	9–15 April	26.96	12.00	71.47	0.00	0.76	24.50	4.43	14.56
16	16–22 April	28.52	13.98	58.23	0.00	0.74	12.00	3.96	13.50
17	23–29 April	24.95	12.14	75.60	0.20	1.21	5.06	3.20	6.81
18	30 April to 6 May	26.49	11.88	69.05	0.00	1.06	3.00	1.83	5.36
Seasonal mean		21.16	8.39	79.77	2.40	0.87	11.42	8.51	13.92
Correlation coefficient (r) between mean aphid population and abiotic factors									
Maximum temperature (°C)							0.52 *	−0.31	0.08
Minimum temperature (°C)							0.11	−0.45	−0.15
Mean relative humidity (%)							−0.41	0.43	−0.04
Total rainfall (mm)							−0.40	−0.09	−0.21
Wind speed (m/s)							−0.53	−0.50 *	−0.53 *

SMW, standard meteorological week; RH, relative humidity; * significant at *p* < 0.05.

**Table 7 insects-12-01080-t007:** Influence of weather factors on mean number of pea aphid, *A. pisum*, per 20 twigs of three lentil varieties during 2017–2018.

		Abiotic Factors					Mean no. of Pea Aphid per 20 Twigs in the Tested Lentil Varieties		
SMW	Date and Month of 2018	Maximum (°C)	Minimum (°C)	Mean RH (%)	Total Rainfall (mm)	Wind Speed (m/s)	Bakria	Abda	Zaria
10	5–11 March	18.36	11.60	94.30	46.80	2.43	6.40	3.23	5.46
11	12–18 March	16.88	8.50	90.37	23.20	1.71	32.51	21.96	25.95
12	19–25 March	15.13	5.66	88.94	11.60	1.21	13.78	12.05	16.23
13	26 March to 1 April	19.81	6.91	86.65	1.60	0.54	32.15	14.86	47.41
14	2–8 April	19.10	8.78	88.88	8.60	0.97	19.96	13.48	19.20
15	9–15 April	15.85	7.72	91.73	55.00	2.04	6.30	4.53	14.66
16	16–22 April	22.84	10.47	86.06	1.80	0.49	4.78	4.60	15.88
17	23–29 April	19.99	11.07	92.30	46.80	0.66	16.25	13.48	32.33
18	30 April to 6 May	19.35	7.68	86.38	4.80	0.67	6.30	4.53	14.65
19	7–13 May	21.20	11.65	86.93	1.00	0.71	4.78	4.60	15.88
20	14–20 May	24.01	11.45	79.19	1.60	0.63	1.95	1.41	2.5
Seasonal mean		19.32	9.23	88.34	18.44	1.10	13.20	8.98	19.10
Correlation Coefficient (r) between Mean Aphid Population and Abiotic Factors									
Maximum temperature (°C)							−0.35	−0.43	−0.14
Minimum temperature (°C)							−0.46	−0.44	−0.39
Mean relative humidity (%)							0.24	0.31	0.18
Total rainfall (mm)							−0.03	0.02	−0.06
Wind speed (m/s)							0.004	0.01	−0.32

SMW, standard meteorological week; RH, relative humidity.

## Data Availability

The raw data used for this manuscript were uploaded to Zenodo under https://doi.org/10.5281/zenodo.4024371 (accessed on 11 September 2020).

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
