# Peer review of "Population Dynamics and Yield Loss Assessment for Pea Aphid, Acyrthosiphon pisum (Harris) (Homoptera: Aphididae), on Lentil in Morocco"

_insects, 2021, doi:10.3390/insects12121080_

Round 1

Reviewer 1 Report

The authors investigated the pea aphid populations and yield losses on three lentil varieties at a single location across three years. They surveyed the peaks of the aphid population throughout each year, and correlated to the population dynamics to possible environmental factors. This information may be useful to be considered for further development of pest management options for the local agencies.

I have some comments that may be helpful to improve the manuscript:  

  1. First, the authors mainly focusing on analyses within each year, however, the aphid population dynamic patterns across three year are obviously different from what is shown in Figure 1. In addition, the aphids were first detected in January 2017, which is about two months earlier than that in the other two years. It will be interesting to know what factors caused the differences across years, and it may be more helpful for people to predict the outbreaks of pea aphids in the future years.
  2. Line 94-104 (Section 2.1): The description of the field experiment setup is not entirely clear, a diagram of the design and/or a picture of the experimental site will be helpful.
  3. Line 107: How the 20 twigs were selected for aphid population survey? And please explain why the two-side rows of the plot were used, my concern is that the edge-effects may take into effects if the edge rows were used, and the population may not fully represented.
  4. The cross pages figures and tables are really hard to follow, please fit the same table/figure onto the same page.
  5. Line 278-281: The authors stated that they observed a consistent pattern, again, I think it will be helpful if you can provide statistical analyses to support this statement (see comment 1).

Reviewer 2 Report

General comments

This paper investigates the impact of pea aphids on lentil yield.  This occurs both directly and indirectly, the latter via transmission of plant pathogenic viruses.  The experiments were run over three years with different varieties of lentils.  In addition, some plots received protection from pesticides in order to calculate avoidable losses in yield.  The paper is straightforward and easy to follow and the results and interpretation reasonable.  The data provide a good first look into what is going on in lentil fields but it does not provide many insights into what drives these patterns.

The paper is largely descriptive and does not dig very deep in attempting to explain the emergent patterns.  There is nothing here that is lentil specific e.g., plant stages for lentil vs fava beans ? (This is briefly mentioned for lentils in the Discussion).  Also, it might be more instructive if the authors had used physiological dates (degree days) for comparing across years.

Unfortunately, the authors do not explain how they chose particular climate variables to explain the population dynamics although all of them seem reasonable.  In addition, predator numbers were not included; why not?  Similarly, though the authors suggested that aphids can directly and indirectly impact lentil yield, there did not seem to be an attempt to relate aphid numbers to yield, per variety.  As noted by the authors, yield loss can be indirect via virus, in which case numbers of aphids might not be a good metric for risk from aphids – perhaps the data can be analyzed in that regard.  Also, are there data from diseased vs healthy plants regarding crop loss and aphid numbers?

All of the questions above point to the fact that this is a good first look at aphid impacts but because they are largely descriptive it is difficult to know how general they are for this pest insect that has already been studied extensively.

Reviewer 3 Report

This study evaluates the population dynamics of the pea aphids associated with lentil during spring in Morocco and estimates the damages to lentil varieties through the infestation. This study also examines to what extent climatic factors determine aphid abundance. The authors analyze the data of the population fluctuations for three years, providing useful information about damages by the pea aphid. However, unfortunately, this study focuses on aphid population dynamics on the local scale only and hardly includes original findings on the biological aspect of the pea aphid, so this paper would attract little attention and interest from the readers. I recommend the authors to submit this manuscript to a regional journal.

Reviewer 4 Report

Fakhouri and colleagues present and discuss the results of a study aiming to investigate pea aphid population dynamics and yield losses on three lentil varieties during three cropping seasons. 

-Although I'm not a native speaker, the English must be seriously improved.

-Line 3: please give the species.

-Line 49: what do you mean "nutritional security". Please expalin.

-Lines 76-68: I can't follow this sentence.

-Lines 77-80, 87-89, 90-91, 316-319 and 334-336: improve flow - reasoning.

Round 2

Reviewer 2 Report

The authors have addressed my concerns.

Author Response

Thank you,

Reviewer 3 Report

First, it is necessary to use the correct scientific name for the pea aphid. Use Acyrthosiphon pisum (Harris). The parenthesis cannot be omitted.

P4, Line 149. The authors conduct Pearson correlation between aphid numbers and meteorological factors. However, because the authors deal with time series data, Pearson correlation between two time series data is not appropriate. Analysis of time series data is not so easy. Even if two parameters follow a random walk process, there will be a seeming trend with time, and you can easily obtain significant correlation between the two parameters. It is necessary to remove this trend to conduct statistical tests.

To test whether the observed fluctuations in aphid numbers tend to show stationarity, it is necessary to conduct the unit root test for the aphid number data using the augmented Dickey–Fuller test (adf.test) (you can use it in R). If stationarity is rejected by the test (i.e. fluctuations follow a random walk process), then the data will produce a seeming trend with time. This result violates the assumption of correlation and regression analyses. Thus, in this case, it is necessary to take difference between neighbouring times (Xt+1-Xt, and Yt+1-Yt) and calculate the correlation between the two differences. If stationarity is accepted by the test, you can use standard correlation analysis.  

In the PDF, I added some corrections.

Author Response

Thank you,
